# Wild Stinging Nettle (*Urtica dioica* L.) Leaves and Roots Chemical Composition and Phenols Extraction

**DOI:** 10.3390/plants12020309

**Published:** 2023-01-09

**Authors:** Živilė Tarasevičienė, Miglė Vitkauskaitė, Aurelija Paulauskienė, Judita Černiauskienė

**Affiliations:** Department of Plant Biology and Food Sciences, Agriculture Academy Vytautas Magnus University, Donelaičio str. 58, 44248 Kaunas, Lithuania

**Keywords:** *Urtica dioica* L., solid–liquid extraction, total phenols, total flavonoids, extraction conditions

## Abstract

Stinging nettle (*Urtica dioica* L.) is an herbaceous plant that grows all over the world and is widely used as an edible and medicinal plant. Overall research results reveal that the chemical content and antioxidant activity of aerial parts and roots of stinging nettle depends on the growing region, soil, meteorological conditions (especially sunshine), collecting time, etc. The chemical composition of stinging nettle growing in Lithuania and the solid–liquid extraction efficiency of leaves and roots using different solvents were analysed. Additionally, we determined leaves phenols extraction efficiency using 96% methanol at different extraction conditions. Research results showed that a higher amount of crude fats, non-nitrogen extractives, and total carotenoids were in leaves, but the amount of crude proteins and ash did not differ significantly compared with roots. A higher amount of polyunsaturated fatty acids (PUFAs) and monounsaturated fatty acids (MUFAs) were detected in roots instead of leaves while saturated fatty acids (SFAs) were in leaves. The extraction results showed that the most effective solvent for total phenols and flavonoids in leaves was 96% methanol, for total phenols in roots was 50% methanol and 50% ethanol for total flavonoids in roots. The most effective temperature for the *Urtica dioica* L. leaves phenols extraction was 70 °C, while time does not have a significant influence. The present study’s findings suggested that concentrated and binary solvents had different effects on the phenol’s extraction efficiency from different stinging nettle parts and extraction temperature performed a key role instead of extraction time.

## 1. Introduction

Stinging nettle (*Urtica dioica* L.) is originally from the colder regions of northern Europe and Asia, but today this herbaceous plant grows all over the world [1]. The plant has been widely used by herbalists around the world for centuries and still is often applied in folk medicine for various disorders. Nettle is known to have anti-inflammatory, antimicrobial, antioxidative, and analgesic effects, boost the immune system, and prevent anemia [2,3]. 

*U. dioica* is a perennial plant with a rich chemical composition. Still, factors such as variety, genotype, climate, soil, vegetative stage, and harvest time affect plants’ nutrients [4]. Research data shows that aerial parts of nettle contain 2.5–3.6% crude fat, 18–34% crude protein, 9% crude fiber, total ash 16%, and 37% carbohydrate in dry matter (DM) [5,6]. The leaves are rich in vitamins C, B, K, and carotenoids such as β carotene, hydroxy β carotene, lutoxanthin, lutein epoxide and violaxanthin [7,8].

Interest in nettles as a medicinal plant has increased significantly in recent years. Scientists are studying the pharmacological effects of the plant, mainly the aerial parts of *U. dioica* that come from flavonol glycosides and phenolic acids [5,9]. Polyphenols are secondary metabolites of plants and are formed as defense against biotic and abiotic stresses. The most abundant phenolic compounds in nettles are rutin, quercetin, 5-O-caffeoylquinic acid, and isoquercetin [10]. Although the aerial parts of the plant, stems, and leaves are mostly used, the nettle roots are rich in phenolic compounds also [9]. The results of a scientific study show that all plant parts (root, stalk, and leaves) have different phenolic compositions and contents [9]. Research results of nettle roots indicated 18 different phenolic components and 8 different lignan components. Other researchers found various phenolic acids in nettle root extracts [9,11]. Investigation data shows that the leaves of stinging nettle accumulate a higher amount of polyphenol than the roots [12]. The total amount of phenolic compounds in nettle roots can vary from 20 to 1020 mg GAE g^−1^ (DM), in stems from 10 up to 480 mg GAE g^−1^ (DM), and in leaves from 150 up to 1941 mg GAE g^−1^ (DM) [9].

The amounts of bioactive compounds in plant material are always fairly low, therefore extracts are often used rather than the raw material [13,14]. Extraction is known as the separation of bioactive compounds from the plant with selective solvents, distillation method, pressing and sublimation leading to the extraction of soluble metabolites from the plant [13]. The amount of biologically active substances in the extract depends on the part of the plant as well as on the harvesting time and the extraction method [4,15,16]. Extracts could be prepared from fresh or dry plant material by using different procedures with solvents such as water, ether, methanol, ethanol, etc., or without solvents (expression, sublimation, and distillation). Extraction with water without boiling is known as an infusion, with boiling as a decoction, repeated extraction with a hot solvent as percolation, and soaking as maceration; also Soxhlet extraction, and pressing of fresh plant material for juice or oil, etc. [17]. The chosen extraction method depends on the raw material properties necessary to obtain a finished product [13,17,18]. Leaf preparations are more suitable for infusion, while hard and woody raw materials such as barks and roots may require decoction or percolation [18]. The bioactive compounds from nettle plants could be extracted using environmentally friendly solvents such as water, and ethanol or non-friendly solvents such as acetone, methanol, etc. [19]. The chemical composition of raw plants and processed products may differ because during processing, some components may be decomposed or their amounts may be decreased, otherwise new components could be formed, changing the ratio of nutrients [18]. The development and optimization of extraction conditions are important for maximizing extraction yield and getting a high-value extract of nettle [15,16,20].

This study aimed to determine the chemical content of different parts of stinging nettle plants growing in Lithuania and phenols extraction efficiency using not-accelerated prolonged-time cold maceration.

## 2. Results and Discussion

### 2.1. Chemical Composition of Different Stinging Nettle (Urtica dioica L.) Parts

The chemical composition of the different stinging nettle plant parts is presented in Table 1. Statistically significant differences were determined only in the amount of crude fats, fiber, and total carotenoid content. The amount of proteins according to the researcher’s data was higher in the nettle leaves than in the roots. Rafajlovska et al. [21] stated that in six different Macedonia regions, collected nettle leaves’ protein content ranged from 16.08 to 26.89% and in roots from 3.31 to 10.89% of dry matter (DM). In our study, the amount of crude fat in leaves was higher than in roots by 1 percent unit and the detected amount corresponds with the Man et al. [22] findings that the amount of fats in nettle leaves reached 2.75%. Additionally, the amount of fiber obtained in their research (8.37%) was close to the amount of fiber in nettle leaves determined in our study (10.58%) (Table 1). Adhikari et al. [6] reported that fiber content in blanched and dried nettle leaves powder was 9.08% of DM. Our research data revealed that fiber content in nettle leaves was almost three times higher than in roots.

The mineral content of analyzed nettle leaves and roots was very close, respectively 18.86 and 18.41%. As was reported in a previous study by Paulauskiene et al. [23] crude ash content depending on collection time ranged from 3.06 to 4.70% of DM with the highest iron content compared to the other elements. Obtained research data was on the line with Adhikari et al. [6] who reported 16.21% of ash in nettle leaves as well as 17.67% in Man et al. [22] research.

Carotenoid accumulation depends on the plant part and on the age of the plant. According to Guil-Guerrero et al. [8] the major carotenoids for all leaf maturity levels were lutein, lutein isomers, β-carotene, and β-carotene isomers. Their research data showed the difference in carotenoid content in young and mature leaves, respectively 5.14 mg 100 g^−1^ and 7.48 mg 100 g^−1^. According to Droštinova et al. [24], the amount of total carotenoid content in conventionally dried nettle leaves was 1.53 mg g^−1^ of DM. Adhikari et al. [6] data showed that carotenoid content in nettle leaves was 3.5 mg g^−1^ of DM. The amount of total carotenoids in the leaves and roots of nettles in our research was significantly different. In leaves, total carotenoid content was 1.2 times higher than in roots, but the overall amount of these compounds was quite low compared to other researchers’ findings (Table 1). It might be the consequence of different growing conditions, soil, plant development stages, exposure of sunlight to plants, as well as other growth factors.

Fatty acids in stinging nettle leaves as well as in roots were determined and the obtained results are presented in Table 2. All detected fatty acids are grouped into saturated fatty acids (SAFs), monounsaturated fatty acids (MUFAs), and polyunsaturated fatty acids (PUFAs). A significant difference was observed between SAFs amount in stinging nettle leaves and roots. SAFs amount in leaves was about 2 times higher than in roots and reached up to 19.05%. Opposite results were obtained in terms of MUFAs and PUFAs. The amount of MUFAs was 1.1 times and PUFAs 1.2 times higher in roots than in leaves. The ratio of SFAs and UFAs (all unsaturated fatty acids) was 4.26 in leaves and 10.21 in roots. According to Durovic et al. [25], this ratio in stinging nettle leaves was 1.12, and PUFAs dominated under the MUFAs by 6.5 times. In our research the distribution of fatty acids was different and the ratio of PUFAs and MUFAs was 1.06 in leaves and 1.16 in roots. High saturated, polyunsaturated, and low monounsaturated fatty acids content was reported in the research of Rutto et al. [26], i.e., 35.5, 61.8, 2.7 and 32.7, 59.8, and 7.5%, respectively when investigating fall and spring nettle leaves.

The most abundant fatty acids in leaves were stearic (C18:0), palmitoleic (C16:1 (n-7)), and linoleic (C18:2 (n-6)), while in roots they were stearic (C18:0), oleic (C18:1 (n-9)), linoleic (C18:2 (n-6)) (Table 2). Other researchers’ data reveals that the most abundant fatty acid in stinging nettle leaves was a-linolenic acid (C18:3 n-3), followed by palmitic acid (C16:0) and cis-9,12-linoleic acid (C18:2 n-6) [8,25]. In our research a-linolenic acid (C18:3 n-3) was not detected in leaves and in the roots was the sparsest of all PUFAs. The most abundant PUFAs linoleic (C18:2 (n-6)) acid content in leaves was 21.93%, in roots 22.20% while tr-linoleic (tr C18:2 (n-6)) acid in roots—15.62% (Table 2). Guil-Guerrero et al. [8] reported the differences in fatty acids in mature and young nettle leaves. According to the researchers, the FAs content in mature leaves was 17.9, 11.6 and 40.7% for C16:0, C18:2, and C18:3, respectively, while in young leaves reported contents were 20.1, 18.1, and 29.6% for C16:0, C18:2, and C18:3 [8]. The researchers analysed the influence of supercritical fluid extraction on the fatty acids content of commercially cultivated and wild-grow stinging nettle. Obtained results revealed that wild-grow plants have synthesized higher amounts of fatty acids than cultivated ones [27,28]. Such a difference in fatty acids composition may be due to the different plant collection times, growing or cultivation conditions as well as fatty acids extraction methods.

Research data shows that the amount of tr-linoleic (tr C18:2) acid of all PUFAs differed the most (Table 2). In the roots, tr C18:2 content was 3 times higher compared to leaves. In the case of MUFAs cis-10-heptadecanoic acid (C17:1), content in the leaves was more than seven times higher than in the roots.

### 2.2. Efficiency of Stinging Nettle (Urtica dioica L.) Polyphenols Extraction 

Extraction conditions have been studied in order to maximize the extraction efficiency of phenolics because results depend on milled plant material particle size and nature of plant matrix, solvent, solid-to-liquid ratio, pH value, extraction time, and temperature [20,29]. One of the major factors affecting the extraction efficiency is solvent polarity; therefore, preliminary experiments were performed in order to select the most effective one for the further extraction optimization process.

The most effective solvent for the solid–liquid stinging nettle leaves extraction was 96% methanol and the least was 96% ethanol (Figure 1). These results were partly in line with the Vajic et al. [20] research results. The authors indicated that the least amount of phenols was extracted in the 96% ethanol and the highest in 50% methanol during the 30 min. of ultrasound-assisted extraction. Additionally, their results suggested that tested methanolic extracts had significantly higher total phenolics content than ethanolic. Their HPLC analyses data revealed that in stinging nettle most abundant phenolics were 2-O-caffeoyl malic acid, chlorogenic acid, and rutin [20]. According to Pinneli et al. [30], 2-O-caffeoyl malic acid and chlorogenic acid in wild and cultivated nettles compose up to 76.5% of total phenolics. 

Our findings showed that after 3 days of the maceration, there were no significant differences between the total amount of phenols in stinging nettle leaves methanol and ethanol extracts of the same concentration, except 96% concentration (Figure 1). The content of phenols fluctuated from 21.60 mg GAE g^−1^ in 96% methanol extract to 11.71 mg GAE g^−1^ in 96% ethanol extract (Figure 1). Methanol is capable to destroy plant cells and to release phenols. Moreover, this solvent is more polar than ethanol and can dissolve polar phenolic compounds such as phenolic acids, glycosides, and plenty of flavonoids. The findings of Tura and Robards [31] presented that methanol also inhibits the activity of polyphenol oxidases, and this can reduce phenolic degradation [31].

Otles and Yalcin indicated that the antioxidant activity of stinging nettle parts is arranged as follows: root > stalk > leaves [9], because the phenols content in the roots differs from that of the aboveground parts [15]. The only prominent compounds in nettle roots were secoisolariciresinol (detected only in root extracts), p-coumaric acid, quinic acid, and scopoletin [15]. In terms of root total phenols content, the most efficient solvents were 50% methanol and 70% ethanol, while the least effective was 96% ethanol. Comparing the highest and the least amount of total phenols in nettle root extracts, the difference was almost 9 times (Figure 2).

According to the results, the total flavonoid content in the nettle leaves extracts was the highest at 96% methanol while the least at 50% methanol. In terms of ethanol, the most effective were 96 and 70% extracts (Figure 3). Water extraction was characterized by higher efficiency than 70 and 50% methanol and 50% ethanol. According to the results, almost all extracted phenols in 96% methanol were composed of flavonoids. Vajic et al. [20] reported that 50% methanol was the most efficient for stinging nettle phenols extraction, but in our research, it was four times less effective than 96% methanol. This may be due to the maceration time and solid-to-liquid ratio. In our study maceration time was prolonged while the solid–liquid ratio was 1.5 times higher (Figure 3).

The total flavonoid content in stinging nettle roots was comparatively lower than in leaves (Figure 4). The highest total flavonoid content was observed in 50% ethanol extract while the least in 96% ethanol. The total flavonoid content in water, 70% ethanol, and 96% methanol was not significantly different. According to Vajic et al. [20], predominant phenol in nettle leaves was rutin, but Jeszka-Skowron et al. [32] research results showed that the lowest content of 3-caffeoylquinic acid or rutin was determined in nettle roots comparing with aerial parts of the plant. In general, the preference of the phenols for methanol and ethanol may be caused by their non-polar part and the aliphatic fragment of alcohols. The bigger molecules prefer ethanol, as it could better “cover” the gaps between the hydrogen bonds while smaller- non-polar solvents,. Extraction yield is enhanced with the increase in the medium polarity [33]. According to Do et al. [34], the extraction yield of phenols from *Limnophila aromatica* was higher in pure methanol than in pure ethanol and pure acetone. This showed that the extraction yield increases with the increasing polarity of the solvent used in extraction. 

### 2.3. Polyphenols Extraction Conditions

The binary solvent system allows one solvent to enhance the solubility of the polyphenols, while the other improves desorption [35]. The quantitative and qualitative polyphenols composition of stinging nettle aerial parts and roots differ. The roots of stinging nettles contain significantly smaller amounts of phenols [15]. Therefore, further research was on stinging nettle leaves. Research data revealed that the most effective solvent for total phenols and flavonoid content extraction in stinging nettle leaves was 96% methanol (Figure 1 and Figure 3). Therefore, for the experiment on the effect of extraction conditions, nettle leaves and methanol were used, despite it being an environmentally hazardous solvent (Table 3). 

Research results indicated that extraction temperature and time for some parameters had a significant influence on extraction efficiency. To evaluate the influence of each factor and the interaction of factors on studied variables, an ANOVA test was carried out with LSD at a 95% confidence level. All factors and their interaction had a significant effect on the extraction efficiency of total phenols and total flavonoids as well as on the antioxidant activity of the extracts, except total phenols content was not dependent on extraction time. According to Silva et al. [29], prolonged extraction time enhances phenols solubility till the moment when the equilibrium of extraction will be achieved. In contrast with that, our research showed that prolonged extraction time was not significant for phenols solubility and was in line with the Vajic et al. [20] and Stevigny et al. [36] findings. The main reasons for the total phenols decrease according to Khoddami et al. [37] might be phenols oxidation and enzymatic degradation under the influence of time. In our research, the most effective extraction conditions for total polyphenols content were extraction at 70 °C for 40 min. With increasing temperature, the amounts of compounds in the extracts and their antioxidant activity increased. The obtained results showed that the highest correlation was between the temperature and total flavonoid content (R = 0.826), followed by correlations between temperature and total phenols (R = 0.608) (Table 4). For extraction of the total content of flavonoids, the most effective conditions were extraction for 60 min. at 70 °C (Table 3). Since the highest content of all flavonoids in nettle was rutin it can be assumed that it is thermostable and 70 °C does not cause degradation. The degradation of rutin according to Chaaban et al. [38] at 70 °C was less than 10% after 2 h of heating while only 50% of the flavonoid content was lost at a temperature of 90 °C. The sharp rutin degradation was observed at a temperature above 100 °C. The antioxidant activity of the extracts was not dependent on extraction time and temperature. (Table 4). The interaction of extraction temperature and time influenced the amount of total phenols and total flavonoids as antioxidant activity of extracts (Table 3).

## 3. Materials and Methods

### 3.1. Materials

All the chemicals used in the food analysis were of analytical grade and obtained from standard commercial suppliers. 

### 3.2. Plant Material

Wild *Urtica dioica* L. plants were collected in the central part of Lithuania (55°25′31.0″ N 22°45′52.5″ E) before blooming in 2020 July (Appendix A). Leaves and roots were separated from the plant and dried in a conventional air dryer Termaks (Kungsbacka, Sweden) at 30 and 40 °C, respectively, for 24 h, and ground with a laboratory ultra-centrifugal mill ZM 200 (Retsch, GmbH, Haan, Germany) to flour consistency (0.2 mm particle size). Samples were stored till the analyses at the −34 °C temperature. The voucher specimen was not prepared during this research.

### 3.3. Determination of Urtica dioica L. Plants Parts Chemical Composition

Crude protein content in stinging nettle was determined by the Kjeldahl method [39], the amount of crude fat by Soxhlet extraction with petrol ether, and the amount of crude ash by burning the sample in muffle L9/11 B180 (Nabertherm, Lilienthal, Germany) at 550 °C. The total amount of crude fiber was detected according to the standardized method [40]. For the analysis of fatty acids composition in stinging nettle gas chromatography methods were used [41,42]. Fatty acids were extracted with n-hexane and saponified with NaOH methanol solution. The fatty acids composition was analysed by using a gas chromatograph GC-2010 Pro (Shimadzu Corporation, Kyoto, Japan) equipped with a BPX–70, 120 m capillary column. Nitrogen was used as a carrier gas. “Supelco 37 Component FAME Mix” (Sigma-Aldrich, Sent Louis, MO, USA) was used as standard for FFA identification, C13:1 tetradecadiene (C14:2) and hexadecadiene (C16:2) fatty acids were identified by interpolation. 

Total carotenoid content was analysed by using a two-ray UVS-2800 spectrophotometer (Labomed Inc., Los Angeles, CA, USA). The absorbance was read at 470 nm and the amount of pigments was calculated [43].

### 3.4. The Extraction Conditions of Phenols and Flavonoids from Urtica dioica L.

Solid–liquid extraction (maceration) was used for the preparation of the extracts. As a solvent, water, ethanol of 50%, 70%, and 96%, and methanol of 50%, 70%, and 96% were used. We briefly weighed 5 ± 0.001 g of ground leaves and roots of *Urtica dioica* L. into separate sealed bottles of dark glass and poured with 150 mL of solvent, stored at 4 °C in the dark for 72 h. The prepared extracts were filtered through 0.2 µm pore size filter paper.

### 3.5. Evaluation of the Efficiency of Extraction Conditions for Urtica dioica L. Leaves

After the extraction of the phenols and flavonoids with different solvents, it was found that the higher amount of phenolic compounds was present in nettle leaves extracts and the most effective solvent for the extraction of phenolic compounds from the leaves were 96% methanol therefore the analysis of extraction conditions efficiency was performed with this solvent and nettle leaves. An amount of 5 g of ground stinging nettle leaves was filled up with 96% methanol with a ratio of 1:30 (leaves: solvent). The extraction was carried out in an ultrasonic laboratory bath AU-65 (Argolab, Carpi, Italy) with an ultrasound frequency of 40 kHz. Extracts were filtered through 0.2 µm pore size filter paper and poured into dark glass vials and stored till the analysis at 4 °C.

### 3.6. Determination of the Chemical Composition of Urtica dioica L. Leaves and Roots Extracts

The total phenolic content was established by using the Folin–Ciocalteu reagent. Briefly, 0.2 mL of the leaves and roots extract was mixed with 5 mL of distilled water, and 0.2 mL of Folin–Ciocalteu reagent. The mixture was left for 6 min at room temperature, 1 mL of 20% Na_2_CO_3_ solution was added to the mixture and left for an hour. Absorbance was measured at a wavelength of 765 nm using a two-ray UVS-2800 spectrophotometer (Labomed Inc., Los Angeles, CA, USA). The results were expressed as mg of gallic acid equivalents (GAE) per gram of dry matter (DM) of plant material) [44].

The total flavonoid content was determined by a method based on Chatatikun et al. [45] and Sandip et al. [46]. In a test tube, 0.2 mL of leaves extract and 1 mL of root extract were mixed with 0.2 mL of AlCl_3_, 3 mL of 96% C_2_H_5_OH, and 0.2 mL of (C_2_H_3_NaO_2_). The prepared mixture was well mixed and left for 40 min at room temperature in the dark. Absorbance was measured at a wavelength of 415 nm using a two-ray UVS-2800 spectrophotometer (Labomed Inc., Los Angeles, CA, USA). The results were expressed as mg of quercetin equivalent (QE) per gram of dry matter (DM) of plant material.

The antioxidant activity of nettle extracts was determined by the 2.2-diphenyl-1-picrylhydrazyl free radical (DPPH) scavenging method [13]. Briefly, 0.3 mL of nettle leaves and roots extracts were diluted with 8.7 mL of methanolic DPPH solution, and samples were shaken for 30 min in the dark. Absorbance was measured at a wavelength of 517 nm using a two-ray UVS-2800 spectrophotometer (Labomed Inc., Los Angeles, CA, USA). DPPH radical scavenging activity was calculated according to the equation:$$\mathrm{DPPH}\;\mathrm{radical}\;\mathrm{scavenging}\;\mathrm{activity}\;\left(\%\right)=\frac{{\mathrm{At}}_{0}-{\mathrm{At}}_{s}}{{\mathrm{At}}_{0}}\times 100;$$where At_0_—absorbance of the control solution containing methanol, and At_s_—absorbance of sample.

### 3.7. Data Statistical Analysis

The data obtained from three replications were analysed by one and two-ways analysis of variance (ANOVA) using Statistica version 12 software (StatSoft, Inc., Tulsa, OK, USA). Statistical reliability between study data was assessed by Fisher’s (LSD) test at the significance level of 0.05. The relationship between measured parameters was assessed by Pearson’s linear correlation test at a *p* < 0.05 significance level. 

## 4. Conclusions

The results suggested that Stinging nettle leaves and roots differed in the context of crude fat, fiber, and non-nitrogen extractives. A significant difference was observed in the content of fatty acids, where PUFAs and MUFAs were more abundant in the roots while SFAs in the leaves. Stinging nettle is a plant of great importance not only in medicine or pharmacology but also in the food industry, etc. Therefore, the most effective extraction of biologically active compounds allows wider and more efficient use of resources. As the study results suggested, cold maceration was an effective extraction method for stinging nettle phenols. Binary solvents were more effective for phenols and flavonoid extraction of *Urtica dioica* L. roots while concentrated of leaves. In terms of extraction conditions, extraction time does not have a significant influence on the extraction efficiency, while the most effective temperature was 70 °C. Future research, which would include *Urtica dioica* L. above ground and underground samples along with chemical analysis of extracts obtained after cold maceration, and extraction using different solvents would contribute to defining the effectiveness of binary and concentrated solvents for phenolic compounds extraction.

## Figures and Tables

**Figure 1 plants-12-00309-f001:**
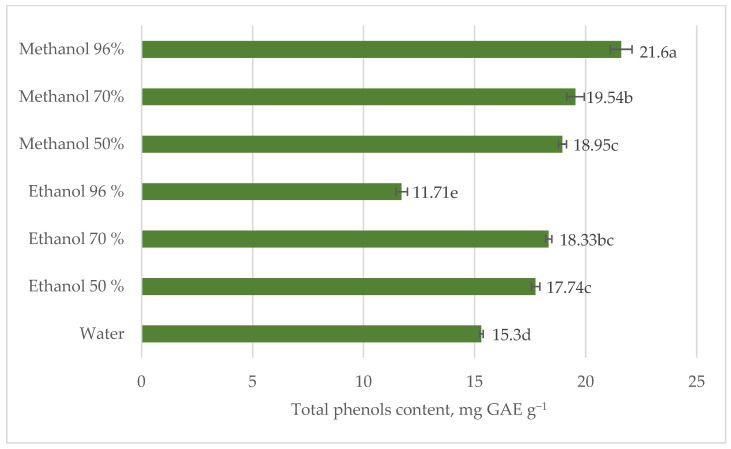
Total phenols content in stinging nettle (*Urtica dioica* L.) leaves extracts, mg GAE g^−1^ DM.

**Figure 2 plants-12-00309-f002:**
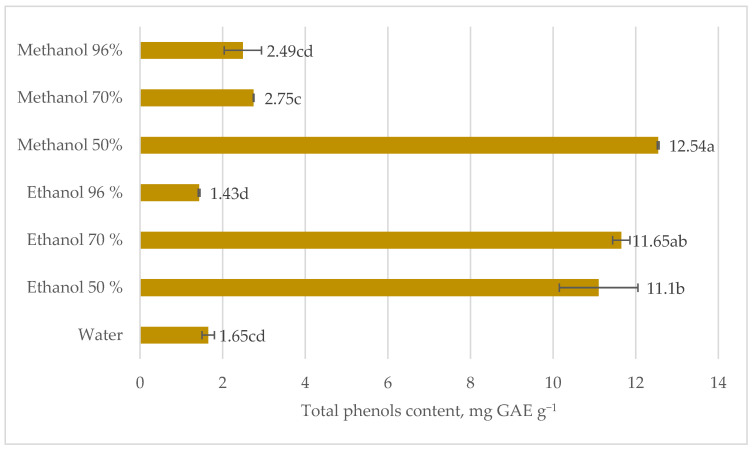
Total phenols content in stinging nettle (*Urtica dioica* L.) roots extracts, mg GAE g^−1^ DM.

**Figure 3 plants-12-00309-f003:**
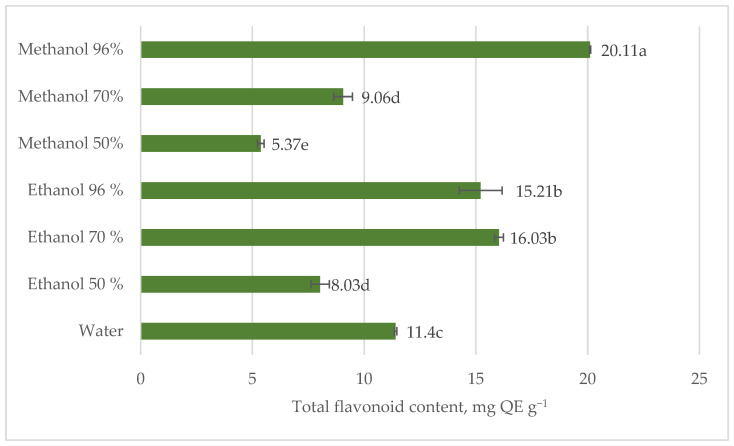
Total flavonoid content in stinging nettle (*Urtica dioica* L.) leaves extracts, mg QE g^−1^ DM.

**Figure 4 plants-12-00309-f004:**
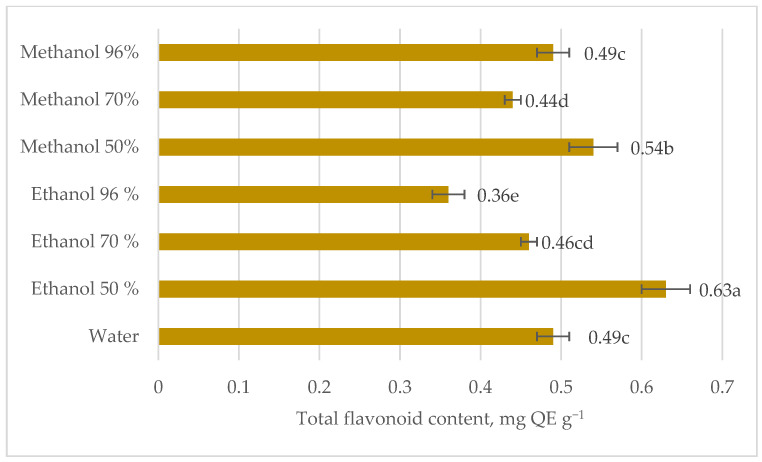
Total flavonoid content in stinging nettle (*Urtica dioica* L.) roots extracts, mg QE g^−1^ DM.

**Table 1 plants-12-00309-t001:** Chemical composition of stinging nettle (*Urtica dioica* L.) leaves and roots (DM).

Chemical Parameters	Leaves	Roots
Crude proteins, %	14.53 ± 0.32 a *	13.66 ± 0.42 a
Crude fats, %	2.62 ± 0.09 a	1.65 ± 0.26 b
Crude fiber, %	10.58 ± 0.08 b	31.65 ± 0.81 a
Crude ash, %	18.86 ± 0.11 a	18.41 ± 0.17 a
Non-nitrogen compounds, %	53.41 ± 0.59 a	34.63 ± 1.14 b
Total carotenoid content, mg 100 g^−1^	2.51 ± 0.07 a	2.17 ± 0.09 b

* Data expressed as means ± standard deviation. Significant differences (*p* < 0.05) between averages in columns are marked by different lowercase letters.

**Table 2 plants-12-00309-t002:** Fatty acids profile of stinging nettle (*Urtica dioica* L.) leaves and roots, % of all fatty acids amount.

Systematic (Trivial) Name	Shorthand Nomenclature	Leaves	Roots
Butanoic (Butyric)	C4:0	0.597 a *	0.112 b
Hexanoic (Caproic)	C6:0	0.482 a	0.082 b
Octanoic (Caprylic)	C8:0	0.473 a	0.096 b
Decanoic (Capric)	C10:0	0.834 a	0.213 b
Dodecanoic (Lauric)	C12:0	1.143 a	0.296 b
Tridecanoic (Tridecylic)	C13:0	0.363 a	0.070 b
Tetradecanoic (Myristic)	C14:0	0.368 a	0.061 b
Pentadecanoic (Pentadecylic)	C15:0	0.614 a	0.205 b
Hexadecanoic (Palmitic)	C16:0	1.087 a	0.391 b
Heptadecanoic (Margaric)	C17:0	0.602 a	0.277 b
Octadecanoic (Stearic)	C18:0	4.120 a	4.166 a
Eicosanoic (Arachidic)	C20:0	1.034 a	0.256 b
Heneicosanoic (Heneicosylic)	C21:0	0.903 a	0.213 b
Docosanoic (Behenic)	C22:0	3.283 a	1.110 b
Tricosanoic (Tricosylic)	C23:0	0.515 a	0.256 b
Tetracosanoic (Lignoceric)	C24:0	2.634 a	0.897 b
**Total amount of SFAs**		**19.052 a**	**8.917 b**
*cis*-9-Tetradecenoic (Myristoleic)	C14:1 (n-5)	2.987 a	0.499 b
*cis*-9-Hexadecenoic (Palmitoleic)	C16:1 (n-7)	12.997 a	10.416 b
*cis*-10-heptadecanoic	C17:1 (n-7)	1.433 a	0.199 b
*cis*-10-pentadecenoic	C15:1 (n-5)	0.911 a	0.185 b
*cis*-9-Octadecenoic (Oleic)	C18:1 (n-9)	10.629 b	16.615 a
tr-9-Octadecenoic (Elaidic)	tr C18:1 (n-9)	10.551 b	13.778 a
Eicosenoic	C20:1	-	0.491 a
**Total amount of MUFAs**		**39.508 b**	**42.183 a**
6,9,12-Octadecatrienoic (g-linolenic)	C18:3 (n-6)	8.474 a	8.823 a
9,12-Octadecadienoic (linoleic)	C18:2 (n-6)	21.926 a	22.203 a
tr-linoleic	tr C18:2 (n-6)	4.822 b	15.615 a
*cis*-5,8,11,14,17-Eicosapentaenoic (Eicosapentaenoic)	C20:5 (n-3)	0.626 a	0.257 b
*cis*-8,11,14-eicosatrienoic (Dihomo-g-linolenic)	C20:3 (n-6)	1.193 a	0.529 b
*cis*-11,14-eicosadienoic (Eicosadienoic)	C20:2 (n-6)	0.819 a	0.488 b
9,12,15-Octadecatrienoic (a-linolenic)	C18:3 (n-3)	-	0.120 a
5,8,11,14-Eicosatetraenoic (Arachidonic)	C20:4 (n-6)	3.580 a	0.855 b
cis-13,16-docosadienoic	C23:2 (n-3)	0.253 a	-
**Total amount of PUFAs**		**41.693 b**	**48.90 a**

* Data expressed as means. Significant differences (*p* < 0.05) between averages in columns are marked by different lowercase letters.

**Table 3 plants-12-00309-t003:** Effect of extraction conditions on total content of phenols, flavonoids, and antioxidant activity of 96% methanol extract of stinging nettle (*Urtica dioica* L.) leaves.

Extraction Temperature, °C	Extraction Time, min	Total Phenols Content, mg GAE g^−1^ DM	Total Flavonoid Content, mg QE g^−1^ DM	DPPH Radicals Scavenging Capacity, %
30	20	20.72 ± 0.14 c*	13.27 ± 2.03 f	53.60 ± 2.38 f
30	40	20.37 ± 0.52 c	17.21 ± 1.47 ef	78.99 ± 1.19 e
30	60	21.09 ± 2.95 c	21.04 ± 0.07 de	85.72 ± 0.22 ab
50	20	23.63 ± 0.36 b	22.50 ± 1.96 cd	84.81 ± 0.21 bc
50	40	17.62 ± 0.25 d	21.08 ± 1.25 de	86.16 ± 0.68 ab
50	60	19.52 ± 0.86 cd	22.86 ± 2.30 cd	86.89 ± 0.16 a
70	20	24.08 ± 1.06 b	26.12 ± 0.69 c	85.910.03 ab
70	40	28.70 ± 2.03 a	38.79 ± 0.04 b	83.35 ± 0.11 cd
70	60	25.15 ± 0.07 b	43.81 ± 1.11 a	83.14 ± 0.35 d
*p-value:*				
Extraction temperature		0.0000	0.0000	0.0000
Extraction time		0.2405	0.0001	0.0000
Interactions of extraction temperature and time		0.0000	0.0009	0.0000

* Data expressed as means ± standard deviation. Significant differences (*p* < 0.05) between averages in columns are marked by different lowercase letters.

**Table 4 plants-12-00309-t004:** Correlation matrix for the relationship for extraction process variables and detected analytes of the extracts.

	t	T	TP	TF	AA
**t**	1.000				
**T**	0.000	1.000			
**TP**	−0.132 ns	0.608 *	1.000		
**TF**	0.373 ns	0.826 *	0.672 *	1.000	
**AA**	0.429 ns	0.465 ns	0.131 ns	0.419 ns	1.000

t—extraction time, T—extraction temperature, TP—total phenols, TF—total flavonoids, AA—antioxidant activity, DPPH radicals scavenging capacity. * Statistically significant at *p* < 0.05; ns—not significant.

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
