# Peer review of "Wild Stinging Nettle (Urtica dioica L.) Leaves and Roots Chemical Composition and Phenols Extraction"

_plants, 2023, doi:10.3390/plants12020309_

Round 1

Reviewer 1 Report

In my opinion, you have done a lot of analysis and collected a lot of useful data. However, some of your results are superficial - for example, you did not take into account the bound fraction of phenolic compounds, despite its considerable amount in Stinging nettle. Moreover, you have used only non-specific methods - for total phenolic or flavonoid content. Therefore, you should think about purifying your extracts (for example, you could use SPE columns to remove most of the interfering substances). Nevertheless, Plants has a IF > 4.5, so determination and identification of phenolic compounds would be mandatory. If you do not have access to LC-MS, please expand your discussion using literature data. Stining nettle is a common and well known plant and there are some good publications detailing phenolic composition. Based on your results, what can you assume about the extraction of these compounds (including flavonoids) in terms of their structure at 96%/70%50% methanol, ethanol polarity? Which of the known phenolic compounds are more termolabile then others (influence of extraction temperature and time)?

In addition, you could also use a different method to determine antioxidant activity, to test the efficacy of your extract against different free radicals, or to be performed in an emulsion instead of a homogeneous medium, etc.

Line 280: Please add to the text which internal standard was used to determine the fatty acid methyl esters. Also, I strongly recommend a picture comparing the chromatograms showing the differences in the fatty acid profile between leaves and roots.

Line 289: Can you please explain to me why you chose 4°C when you tested different extraction solvents? Is not this temperature too low? And without any stirring, ultrasonication, etc. that would accelerate mass transport?

Line 328: Why did you choose Fisher's LSD post-hoc test?

Author Response

Answers to reviewer 1

Thank you very much for your comments and suggestions that will improve the article.

The 2,2-diphenylpicrylhydrazyl (DPPH) assay was used to assess the ability of phenolic compounds to transfer labile H atoms to radicals. This method is common in plant biochemistry to evaluate the properties of plant constituents, mainly polyphenols for scavenging free radicals. Very often, this method is used as the main and only one, without supplementing with other methods for the determination of phenols' antioxidant activity (DOI: 10.1016/j.indcrop.2015.06.032; doi.org/10.1016/j.jarmap.2019.100229; doi:10.1100/2012/564367; etc.).

Line 289: Maceration is still commonly used since they only require basic glassware or steel containers and are convenient for both small and large-scale extraction. However, maceration is time and solvent-consuming and maceration acceleration by heating is not suitable for thermosensitive compounds. During the cold maceration (at room temperature or 10 °C) the rationale is to obtain an increased phenolic extraction especially in an aqueous media and at low temperatures to prevent the start of fermentation. 

There were no data about the phenols content after not accelerated with prolonged time cold maceration (4 °C) of Urtica dioica L. The main factor for the transport of the compound is time and mass: solvent ratio. Also high temperatures might cause the destruction of thermolabile components and solvent evaporation.

Line 328. Fisher's LSD method is used in ANOVA to create confidence intervals for all pairwise differences between factor level means (while controlling the individual error rate to a significance level you specify). LSD has more power compared to other post-hoc comparison methods (e.g., the honestly significant difference test, or Tukey test) because the α level for each comparison is not corrected for multiple comparisons (https://personal.utdallas.edu/~herve/abdi-LSD2010-pretty.pdf).

Reviewer 2 Report

This interesting work drawn up clearly the new possibilities of using of Urtica dioica L. leaves and roots due to the different extracts and their concentrations.

However there are some major concerns regarding the quality of this work:

The title is unfortunately misleading. It is not a chemical composition, but only the total content of phenols and flavonoids. Therefore, the title should be reworded to reflect the actual content of the article and the experiments carried out. There is no information in the entire study which polyphenolic substances were identified as a result of the HPLC analysis of extracts. It would be good if the authors provided information on what polyphenols were dominant in these extracts and how the extracts obtained from different parts of plants differed. After all, they used different solvents (see paragraph 305-323).

Thus it is necessary an additional chemical characterisation of these extracts regarding the identification of the components from different types of extracts. The total phenolic content measured by Folin method is an important information, but in some cases it is difficult to correlate with antioxidative properties as phenolics vary in their activity. The second aspect to have in view is that plant phenols have antioxidant properties as well as the potential to act as prooxidants under certain conditions.

Line 265-270: You must name the botanist who identified this plant and in which Herbie you deposited a voucher specimen of the plant.

Line 310-311: it is not clear the expression of the results of the total phenolic content „mg GAE g-1). The results are found to be expressed in the literature as g of gallic acid equivalents (GAE) per 100 g of extract.

Line 319-323: specify what was the positive control. It is also important to calculate the EC50 values of different extracts as scavengers of DPPH free radical.

Line 418: only at reference 33 is added the article DOI. Please adjust the references using the guide information provided.

Author Response

Thank you very much for your comments and suggestions that will improve the article.

As for the title, we wouldn‘t like to agree that it is misleading because it is not only about the total content of phenols and flavonoids. The chemical composition is presented in Lines 79 – 115 of the article.

The extracts composition was not identified in this study, but in the part of the discussion it is presented the data obtained by other researchers about the composition of the extracts.

Lines 285 – 301 present how were obtained extracts with different solvents from different parts of Urtica dioica L., as well as evaluation of the efficiency of extraction conditions for Urtica dioica L. leaves using 96% methanol.

Line 265-270. Urtica dioica L is very common in Lithuania and the identification of this plant is not in doubt. Anyway, the photos were checked in plant.ID (https://plant.id/; https://cordis.europa.eu/article/id/436445-plant-identification-making-the-unknown-known).

Line 310-311. Specified.

Line 319-323. DPPH analysis specified. EC50 shows the concentration (or dose) effective in producing 50% of the maximal response and is a convenient way of comparing drug potencies. This is more important in a pharmacological context.

Line 418. The DOI included to all references in the list.

Reviewer 3 Report

The aim of this paper was to investigate the chemical content determination of different parts of stinging nettle plants growing in Lithuania. The manuscript is interesting and fits within the scope of the journal. The title is clear and adequate to the article’s content. Some revisions are necessary to improve the clarity of the presentation:

-          There are significant grammatical mistakes and long sentences that make the sentence lose meaning and difficult to understand. There are a few spelling mistakes, so improve the English language also.

-          In the abstract, please include an explanation for the abbreviation.

-          Please include a general conclusion in the abstract.

-          Please include the voucher number for plant material.

-          Please change ml in mL (ex: L289, 306, 308, 314….)

-          Please rewrite the conclusions so that the presented results are not repeated. Here the authors must discuss general results and especially future research directions. So, include in the text potential research directions. What are the future applications? What are the next research directions?

Author Response

Thank you very much for your comments and suggestions that allowed to improve the article.

Round 2

Reviewer 1 Report

You have successfully considered my comments and expanded the discussion accordingly. I have no further comments on the manuscript as it stands. However, to justify the publication of your article in the Plants, the quality of your work should be improved. The extracts obtained with different solvents should be analysed  by qualitative and quantitative methods, because only on the basis of their composition data could you draw credible conclusions from your study.

Author Response

Dear Reviewer.

Thank you very much for your comments and suggestions.

Reviewer 2 Report

I observed that authors didn't respond targeted to the requests. I encourage to resubmit the manuscript taking into consideration the following aspects.

"The extracts composition was not identified in this study, but in the part of the discussion it is presented the data obtained by other researchers about the composition of the extracts."

Because it is considered to be an original work and not a review, it is important that reporting to the litterature to be used in the discussion section for demonstrating why or how their work is similar or different from prior studies and NOT ONLY to present the results that were previously obtained. Secondly, the chemical composition of plant material may differ from a region to another (even it may differ from year to year in the same area).

So, this information which the authors rely on is quite relative and the results can be sometimes unreproducible, depending on natural conditions. That’s why it is important to chemically characterize the extracts they have obtained and to present a real information regarding them.

„Line 265-270. Urtica dioica L is very common in Lithuania and the identification of this plant is not in doubt. Anyway, the photos were checked in plant.ID (https://plant.id/; https://cordis.europa.eu/article/id/436445-plant-identification-making-the-unknown-known).”

I agree with the fact that the plant is common in Lithuania. Unfortunately, due to the existence of 46 species of the genus Urtica and also of different varieties of Urtica dioica it is quite important that the identification of the plant to be done by a specialist botanist and not only by a photo analysis. See the following reference: https://www.ncbi.nlm.nih.gov/pmc/articles/PMC6100552/

Also, the voucher specimen of the plant should be preserved for future need, if it will be necessary to do again the extracts.

„Line 319-323. DPPH analysis specified.”

I didn’t observed the information regarding the positive control that was used in the article. It is not good to report to a result found in litterature. It is important to have a positive control when the authors determine the antioxidant scavenging activity. Thus the obtained results can be compared with the results of positive control to see if the extracts are active or not as DPPH scavengers.

Having in view that the authors obtained different types of extracts, EC50 value calculation is very important to express the antioxidant capacity and to compare their activity. It represents the concentration required to obtain a 50% antioxidant effect and it is used not only in pharmacological tests but in antioxidant tests. So, I encourage the authors to calculate this parameter for a better discussion regarding the influence of solvents on the extracts.

In pharmacological context it is used ED50 value (efficient dose 50).

Author Response

(The authors gave the same response as above.)

Reviewer 3 Report

The authors revised largely in agreement with my comments. But they did not answer in a cover letter point by point. Pay attention to the scientific name of the species - Urtica dioica L. not Stinging nettle L., as it appears in the abstract. The voucher number was not entered. Please revise the English language.

Author Response

(The authors gave the same response as above.)

Round 3

Reviewer 2 Report

Dear authors. Thank you for your answers. Please add the new information provided in your last answers in the main text also.

Author Response

Dear reviewer.

Thank you for your valuable comments which helped to improve the quality of the article.